# Extended Warranty Strategy and Its Environment Impact of Remanufactured Supply Chain

**DOI:** 10.3390/ijerph19031526

**Published:** 2022-01-28

**Authors:** Xuemei Zhang, Jiawei Hu, Suqin Sun, Guohu Qi

**Affiliations:** 1School of Business, Fuyang Normal University, Fuyang 236037, China; xmz@mail.ustc.edu.cn (X.Z.); hujiawei@stu.fynu.edu.cn (J.H.); sunsuqin@stu.fynu.edu.cn (S.S.); 2Anhui Provincial Key Laboratory of Regional Logistics Planning and Modern Logistics Engineering, Fuyang Normal University, Fuyang 236037, China

**Keywords:** remanufactured supply chain, e-commerce platform, extended warranty service, trade old for new, trade old for remanufactured, environment impact

## Abstract

To reduce environmental pollution, the government has issued relevant laws and regulations, and more and more enterprises engage in remanufacturing and recycling used products. Trade old for new and trade old for remanufactured have become marketing means to promote product recycling. The extended warranty service is used to promote the recycling of waste products. To design an optimal extended warranty service strategy and analyze its environment impact in a remanufactured supply chain, game theory is used to model the competitive relationship between a manufacturer and an E-commerce platform. Considering whether the E-commerce platform provides extended warranty service, four models are constructed, and the extended warranty service strategy and its environment impact can be analyzed. The results show that, when the level of substitutability between remanufactured and new products meets a certain rage, new or remanufactured products with extended warranty service strategy can increase the demand for new or remanufactured products, respectively. In the four models, the changing trends of manufacturer’s profit, E-commerce platform’s profit and supply chain’s profit, consumer surplus, environmental impact and social welfare are the same, but only the thresholds are different. From the perspectives of supply chain member, supply chain system, consumer, environment and society, the new and remanufactured products with extended warranty service strategy is the best choice.

## 1. Introduction

With the rapid development of technology, the update speed of electronic products is faster than ever before; consumers’ pursuit of fashion creates the average lifespan of electronic products, such as mobile phones from 2.9 years old in 2011 to 2.21 years old in 2018 [1]. Estimates show that approximately 54 million tons of waste electronical equipment were generated globally in 2019. In addition, climate change has led more countries, such as Australia and Sweden, introducing relevant policies to pay attention to climate change [2,3]. Germany, Norway, the Netherlands and other countries and regions have proposed a zero emission timetable [4]. The development of circular economy and sharing economy to promote environmental protection and rational use of resources has attracted the attention of the whole world [5,6,7]. The recycling of resources is an important means to develop a circular economy and sharing economy, and it is also an effective measure to protect the environment [8]. The recycling and remanufacturing of products can not only extend the life of products and realize the efficient recycling of resources, but also, the production cost is about 50% of the new product, energy-saving is 60%, material-saving is more than 70% and pollution to the atmosphere and water resources is reduced nearly 80%, resulting in a remanufactured supply chain (RSC) [9,10]. It can achieve the effect of reducing costs and protecting the environment at the same time [11,12,13].

In the circular economy environment, many countries have introduced relevant policies to promote recycling and remanufacturing [14,15,16]. For example, in May 2020, the National Development and Reform Commission, the Ministry of Industry and Information Technology and other seven departments issued the “Implementation Plan on Improving the Recycling and Treatment System of Waste Household Appliances and Promoting the Renewal of Household Appliances Consumption” to renewal consumption activities such as the trade-in of home appliances in urban and rural areas (https://www.ndrc.gov.cn/xxgk/zcfb/tz/202005/t20200515_1228206_ext.html, accessed on 14 May 2020). After that, many companies began to trade in the old trade for new. For instance, in order to further encourage ISO users to switch to Android, Samsung provides shoppers with a Samsung credit line of up to $200 and a trade in transaction of up to $700. Amazon announced free Galaxy buses and duo pad with the Galaxy S20 5G device (http://www.techxue.com/xinwen/202002/11535.html, accessed on 22 February 2020). In August 2013, Apple launched the iPhone trade old for new project for the first time and then expanded it to iPads and Macs (https://www.chinaz.com/news/2015/0331/394768.shtml, accessed on 11 March 2015). Domestic brands such as Lenovo have also begun to launch online trade-in services. Surface proposed to trade Pro 8 for new: enjoy an additional subsidy of up to ¥600 to support “purchase first and then recycle” (https://www.microsoftstore.com.cn/tradein, accessed on 24 January 2022). Lululemon, a famous Yoga sportswear retailer in Canada, announced that it would pilot the “trade-in program” in California and Texas starting from May 2021 (https://baijiahao.baidu.com/s?id=1698078151935716584&wfr=spider&for=pc, accessed on 26 April 2021). In the whole year of 2020, Suning Tesco collected 3.15 million home appliance recycling orders and 2.34 million replacement appliances, with a total conversion amount of nearly 7 billion [17].

With the development and progress of manufacturing technology, the performance and quality of remanufactured products are no less than those of new products [18,19]. However, due to market reasons such as consumer perception and preference, remanufactured products currently perform poorly in the terminal sales link, which severely restricts the future development of the remanufacturing industry. A number of studies have shown that the length of the warranty has a significant impact on the price of remanufactured products, so warranty strategies have also been used to increase consumers’ willingness to buy remanufactured products [20,21,22]. In order to promote the future development of the recycling and remanufacturing industry, more companies choose to provide good warranty services to reduce the impact of consumers’ cognitive preferences [23,24,25,26]. For example, JD.com displays different extended warranty services for consumers to choose from (https://www.jd.com/pinpai/982-140289.html, accessed on 24 January 2022). Yuan Weijie, the person in charge of Meiyan Warranty Greater China, said, “After three of four years of cooperation with us, although an extended warranty service needs to increase the price of the car by 2–3%, the return rate of maintenance is high, locking the customer, improving the customer’s return rate, and this part of the money can be earned through later parts sales” (https://www.sohu.com/a/229569271_115706, accessed on 26 April 2018). More E-commerce platforms have set up different extended warranty service options in their product purchase interfaces. These services increase consumers’ willingness to buy and promote the good operation of the supply chain’s forward sales and reverse recycling [27]. Then, companies can achieve economic and environmental goals by offering extended warranties.

Consumers can get new products or remanufactured products at discounted prices from E-commerce platforms by “trade old for new” (TON) or “trade old for remanufactured” (TOR) [28]. Meanwhile, the E-commerce platform provides consumers with different extended warranty services (EWS). Different EWS will inevitably affect the profits of manufacturers and E-commerce platforms and social welfare [26]. When TON and TOR exist at the same time, this paper investigates the EWS strategy of the remanufactured supply chain. The purpose of this paper is solving the following three problems:How do the factors such as consumers’ sensitivity to the EWS length affect the demands of new and remanufactured products, the profits of supply chain members and the remanufactured supply chain system, consumer surplus, environment impact and social welfare with the coexistence of TON and TOR?How will different EWS strategies affect the benefits of supply chain members, the remanufactured supply chain system, consumers, environment and society?Which EWS strategy is the best for supply chain members, the supply chain system, consumers, environment and society?

The remainder of this paper is organized as follows. Section 2 reviews the related literature. Section 3 describes the problem and model assumptions. In Section 4, the Stackelberg equilibrium results of the four models are structured. Section 5 analyzes the equilibrium results of four models. Section 6 illustrates the numerical analysis. The conclusions are summarized in Section 7.

## 2. Literature Review

The literature related to this paper mainly includes three aspects: remanufactured supply chain management, the recycling strategy of TON or TOR and EWS strategy.

### 2.1. Remanufatured Supply Chain Management

Scholars have paid more attention to the research of circular economy and remanufactured supply chain management [29,30,31,32]. One focus is on the selection and design of the mode of RSC. Huang et al. [33] and Liu et al. [34] constructed the dual recycling channels of a closed-loop supply chain and studied which mode was the best. Zhang et al. [35] studied the selection of third-party remanufacturing models for closed-loop supply chains with capital constraints under financing portfolios. Secondly, another focus is the decision of pricing. Liu et al. [36] investigated how retailers’ fairness concerns affect cooperative relationships in a three-party sustainable supply chain. Zheng et al. [16] considered a remanufacturing supply chain and analyzed the financing decisions for the remanufacturing supply chain in terms of the market uncertainty. Kabul et al. [24] constructed a decentralized supply chain that consisted of a retailer and a supplier and investigated the pricing decision of the retailer and the supplier, respectively. Thirdly, a focus is on the choices in remanufactured mode. Giri and Clock [37] examined the bullwhip effect in a manufacturing/remanufacturing supply chain. Tan et al. [38] studied the impact of the manufacturer’s two remanufacturing models on the supply chain, and the results showed that, for a market with obvious green consumerism, self-remanufacturing is more beneficial. Li et al. [39] constructed a game theory model between a manufacturer and a remanufacturer under government subsidies and carbon tax policies. Luo et al. [40] developed four game theory models to evaluate the impact of the carbon tax policy on manufacturing and remanufacturing decisions in a closed-loop supply chain. When new and remanufactured products coexist, consumers’ willingness to buy new and remanufactured products has attracted extensive attention [41,42,43]. Gan et al. [44] developed a pricing decision model for short life cycle products in a closed-loop supply chain. Hazen et al. [45] examined the ambiguity tolerance, perceived quality and willingness to pay for remanufactured products and found evidence to support a direct relationship between a consumer’s tolerance for ambiguity and their willingness to pay for remanufactured products. Most of the existing literature has studied the pricing decisions of the remanufactured supply chain. Few studies have considered the policy of trade in during the recycling process and extended warranty services and their environment impact.

### 2.2. Recycling Strategy of TON or TOR

Solutions related to the reduction of carbon emissions have acquired importance [3,15,46]. Yang et al. [47] used three paths through which OFDI can affect the carbon dioxide emissions of a whole country, including economic scale, technology level and industry composition effects. Mostafaeipour et al. [48] employed a system dynamics approach to examine the effects of renewable energy development in Iran in terms of carbon emissions and examined five distinct scenarios: increasing the feed-in tariff, eliminating fossil fuel power plant subsidies, gradually eliminating fossil fuel power plant subsidies and two combined scenarios that consider the carbon cost of electricity generation. As a way to reduce carbon emissions, recycling and remanufacturing can not only reduce environmental damage but also obtain cost advantages [49]. Different recycling strategies have different impacts on enterprises, and the choices of recycling strategies of enterprises have also attracted the attention of scholars. Some scholars have studied the pricing decisions of trade-ins. For example, Cao et al. [50] constructed three trade-in recycling channels adopted by retailers and studied the optimal product pricing and trade-in rebates under the three channel choices. Cao et al. [51] explored the optimal recycling strategy and related pricing decisions of the B2C platform with a dual-format retail model. Some scholars have studied the issue of the model of trade-ins. For example, Ma et al. [28] showed that adopting two kinds of trade-ins simultaneously does not necessarily benefit the firm and that the firm should use different trade-in schemes under different conditions. Zhao et al. [52] developed game models considering a duopoly situation where an original equipment manufacturer offers a trade-in program to collect used products and faces competition from a third-party remanufacturer. Ma et al. [53] analyzed five models and developed equilibrium solutions thereof to understand the impacts of double reference parameters and government incentives on pricing strategies, the manufacturer’s profits and the consumer surplus. Xiao et al. [54] studied the optimal discounted prices and resale prices of refurbished products for companies that provide hybrid trade-in programs, and the results showed that hybrid trade-in plans can generate more profits. Tang et al. [55] studied the decision-making of a supplier who adopted TON. The results showed that TON can promote new product sales. The existing literature mostly considered one of the two recycling strategies: trade old for new or trade old for remanufactured, but less considered the coexistence of the two recycling strategies and did not consider the issue of EWS strategy and its environment impact.

### 2.3. EWS Strategy

Researching on the opposite of EWS strategy, one focus is on the pricing of EWS. Lei et al. [56] studied multistage product and warranty dynamic pricing strategies where consumers can understand the product quality. Considering the impact of consumers’ purchasing decisions of extended warranties on warranty costs, He et al. [57] established a warranty cost model based on the product failure processes. Zheng et al. [58] studied the best pricing decisions for flexible extended warranty periods and pointed out that EWSs can be sold at the end of product sales or the end of the basic warranty. Dan et al. [59] considered a dual-channel supply chain in which both manufacturers and retailers can sell homogeneous products bundled with extended warranty services provided by manufacturers. Bian et al. [60] studied the issue of retailers providing additional trade-in services during the warranty period and found that the EWS policy in the form of a trade-in is better than the traditional EWS policy. Huang et al. [61] studied two warranty service provision strategies, i.e., outsourcing to third-party service providers or not outsourcing, for two competing manufacturers, by considering the competition intensity, service costs and product reliability. Jin and Zhou [62] investigated whether remanufactured products in a closed-loop supply chain should have the same warranty policy as new products. The results showed that the same warranty policy is beneficial to the environment. Liu et al. [63] investigated the optimal replacement problem for a warranty product subject to (*M* + 1) types of mutually exclusive failure modes, including *M* repairable failure modes and a catastrophic failure mode. Hosseini-Motlagh et al. [64] found that offering longer warranties for green products could improve the environmental performance of supply chains. The existing literature on EWS has mostly studied the impacts of different entities providing EWS on supply chain decision-making and its impact on the environment but have not studied the specific EWS strategy of the remanufactured supply chain.

## 3. Problem Description and Model Assumptions

### 3.1. Problem Description

We considered an RSC consisting of a manufacturer (M) and an E-commerce platform (E). In the forward channel, M wholesales new and remanufactured products to E. Then, E sells new and remanufactured products to consumers through two ways, including TON and TOR. In the reverse channel, E recovers the used products through TON or TOR and sells the used products to M. M models as the leader, followed by E, and they play the dynamic Stackelberg game.

In this RSC, we consider three EWS strategies: cases of new products with EWS (strategy *N*), remanufactured products with EWS (strategy *R*) and new and remanufactured products with EWS (strategy *NR*). The case without EWS is modeled as the benchmark model. Considering the impact of different EWS strategies on the decisions of RSC, four models are constructed: benchmark model without EWS (model B), model under strategy *N* (model N), model under strategy *R* (model R) and model under strategy *NR* (model NR), which are shown in the following Figure 1.

### 3.2. Assumptions and Parameters

Based on the above problem description, we employ the symbols and notations given in Table 1 throughout this paper.

To make the analysis tractable, we introduced the following assumptions.

**Consumer preference.** The market size is assumed to be 1, which does not affect the results [11,65]. The proportion of consumers who prefer new products is a, then the proportion of consumers who prefer remanufactured products is 1−a, 0<α<1 [66,67]. The demand is assumed to be a linear function of price [66], and thus, the demands of new and remanufactured products are $D_{n}=a-p_{n}+bp_{r}$ and $D_{r}=1-a-p_{r}+bp_{n}$, 0<b<1, respectively [68,69,70]. In addition, the demand of EWS is assumed to be a linear function of the service price and service length [71], i.e., $D_{e}=D_{n}+D_{r}-p_{e}+kt$, 0<k<1 and 0<t<1 [60].

**Cost structure.** It is assumed that new products and remanufactured products are the same quality [11,72]. Due to the recycling of used components, producing a remanufactured product is cheaper than producing a new product, i.e., $c_{n}>c_{r}>0$, $\Delta =c_{n}-c_{r}>0$ [34,73,74]. The cost of providing EWS for E is a convex function of the EWS length, and it is formulated as $t^{2}$ [71,75].

**Environmental impact.** In the RSC, the main sources of carbon emission and environmental pollution are the producing and remanufacturing process [11]. It is assumed that the carbon emission generated in producing a unit new product is higher than that in producing a unit remanufactured product, i.e., $e_{n}>e_{r}>0$ [11,72]. The environmental impact is assumed to be linearly increasing in its production output, which is formulated as $E=e_{n}D_{n}+e_{r}D_{r}$ [24,34,71]. The total environment cost is assumed to be vE [11,72].

**Game specification.** In the RSC, a Stackelberg game is modeled between M and E, where M is the leader and E is the follower. Firstly, M determines the wholesale prices of new and remanufactured products and the recycling price, And then, E sets retail prices of new and remanufactured products and the price of EWS.

## 4. Model Formulations

In this part, we derive the equilibrium results for the four models B, N, R and NR. In the following calculation process, for analytical purposes, we denote them as $\phi_{1}=A+c_{n}$, $\phi_{2}=A+c_{r}$, $\phi_{3}=a+b-ab$ and $\phi_{4}=1-a+ab$, and according to the above hypothesis, $\phi_{1}$, $\phi_{2}$, $\phi_{3}$ and $\phi_{4}$ are all positive.

### 4.1. Strategy without EWS (Model B)

In model B, E does not provide EWS to consumers. M wholesales new and remanufactured products to E, and then, E sells these products to consumers by two ways: TON and TOR. M and E play a dynamic Stackelberg game. M firstly determines wholesale prices of new and remanufactured products, and then, E determines the retail prices of new and remanufactured products. The optimization models of profit maximization for M and E are formulated as:(1)$$\begin{array}{l} \underset{w_{n}^{B},w_{r}^{B}}{\max}\pi_{M}^{B}=(w_{n}^{B}-c_{n})D_{n}^{B}+(w_{r}^{B}-c_{r})D_{r}^{B}-A(D_{N}^{B}+D_{r}^{B})\\ s.t.\underset{p_{n}^{B},p_{r}^{B}}{\max}\pi_{E}^{B}=(p_{n}^{B}-w_{n}^{B})D_{n}^{B}+(p_{r}^{B}-w_{r}^{B})D_{r}^{B} \end{array}$$

By using reverse induction, we obtain the optimal results for model B, which are described in the following Theorem 1.

**Theorem** **1.***In model B, the optimal prices are given as *$w_{n}^{B*}=\frac{\phi_{1}}{2}+\frac{\phi_{3}}{2(1-b^{2})}$, $w_{r}^{B*}=\frac{\phi_{2}}{2}+\frac{\phi_{4}}{2(1-b^{2})}$, $p_{n}^{B*}=\frac{\phi_{1}}{4}+\frac{3\phi_{3}}{4(1-b^{2})}$*and* $p_{r}^{B*}=\frac{\phi_{2}}{4}+\frac{3\phi_{4}}{4(1-b^{2})}$.

**Proof.** See Appendix A. 

Thus, the optimal demands for new and remanufactured products are given as $D_{n}^{B*}=\frac{b\phi_{2}}{4}-\frac{\phi_{1}}{4}+\frac{a}{4}$, $D_{r}^{B*}=\frac{b\phi_{1}}{4}-\frac{\phi_{2}}{4}+\frac{1-a}{4}$ and $D^{B*}=\frac{1}{4}-\frac{\left(1-b)(\phi_{1}+\phi_{2}\right)}{4}$.

The optimal profits of M, E and RSC are shown as $\pi_{M}^{B*}=(w_{n}^{B*}-c_{n})D_{n}^{B*}+(w_{r}^{B*}-c_{r})D_{r}^{B*}-A(D_{n}^{B*}+D_{r}^{B*})$, $\pi_{E}^{B*}=(p_{n}^{B*}-w_{n}^{B*})D_{n}^{B*}+(p_{r}^{B*}-w^{B*})D_{r}^{B*}$ and $\pi_{T}^{B*}=\pi_{M}^{B*}+\pi_{E}^{B*}$. The environment impact is given as $E^{B*}=e_{n}D_{n}^{B*}+e_{r}D_{r}^{B*}$.

### 4.2. Strategy N (Model N)

In model N, E provides TON with EWS to consumers. M wholesales new and remanufactured products to E, and then, E sells these products to consumers by two ways: TON and TOR. M and E play a dynamic Stackelberg game. M firstly determines the wholesale prices of new and remanufactured products, and then, E determines the retail prices of new and remanufactured products and the retail price of EWS. The optimization models of profit maximization for M and E are formulated as
(2)$$\begin{array}{l} \underset{w_{n}^{N},w_{r}^{N}}{\max}\pi_{M}^{N}=(w_{n}^{N}-c_{n})D_{n}^{N}+(w_{r}^{N}-c_{r})D_{r}^{N}-A(D_{n}^{N}+D_{r}^{N})\\ s.t.\underset{p_{n}^{N},p_{r}^{N},p_{e}^{N}}{\max}\pi_{E}^{N}=(p_{n}^{N}-w_{n}^{N})D_{n}^{N}+(p_{r}^{N}-w_{r}^{N})D_{r}^{N}+(p_{e}^{N}-t^{2})D_{e}^{N} \end{array}$$

By using reverse induction, we obtain the optimal results for model N, which are described in the following Theorem 2.

**Theorem** **2.***In model N, the optimal prices are given as*$w_{n}^{N*}=\frac{\phi_{1}}{2}+\frac{\phi_{3}}{2(1-b^{2})}+\frac{(k-t)t}{4}$, $w_{r}^{N*}=\frac{\phi_{2}}{2}+\frac{\phi_{4}}{2(1-b^{2})}$, $p_{n}^{N*}=\frac{3\phi_{1}}{4}-\frac{b\phi_{2}}{12}-\frac{3\phi_{3}}{4}+\frac{b\phi_{4}}{12}+\frac{t(t-k)}{6}$, $p_{r}^{N*}=\frac{\phi_{2}}{4}+\frac{3\phi_{4}}{4(1-b^{2})}$*and* $p_{e}^{N*}=\frac{b\phi_{2}}{6}-\frac{\phi_{1}}{6}+\frac{2a+5t^{2}+7tk}{12}$.

**Proof.** See Appendix B. 

Thus, the optimal demands for new and remanufactured products are shown as $D_{n}^{N*}=\frac{b\phi_{2}}{3}-\frac{\phi_{1}}{3}+\frac{2a+kt-t^{2}}{6}$, $D_{r}^{N*}=\frac{b\phi_{1}}{3}-\frac{(3+b^{2})\phi_{2}}{12}+\frac{\phi_{4}}{4}+\frac{b(t^{2}-tk-2a)}{6}$, $D^{N*}=\frac{(1-b)(b-3)\phi_{2}}{12}-\frac{(1-b)\phi_{1}}{3}+\frac{\phi_{3}}{12}+\frac{3-b+2t(1-b)(k-t)}{12}$ and $D_{e}^{N*}=\frac{b\phi_{2}}{6}-\frac{\phi_{1}}{6}+\frac{2a-7t(t-k)}{12}$.

The optimal profits of M, E and RSC are shown as $\pi_{M}^{N*}=(w_{n}^{N*}-c_{n})D_{n}^{N*}+(w_{r}^{N*}-c_{r})D_{r}^{N*}-A(D_{n}^{N*}+D_{r}^{N*})$, $\pi_{E}^{N*}=(p_{n}^{N*}-w_{n}^{N*})D_{n}^{N*}+(p_{r}^{N*}-w_{r}^{N*})D_{r}^{N*}+(p_{e}^{N*}-t^{2})D_{e}^{N*}$ and $\pi_{T}^{N*}=\pi_{M}^{N*}+\pi_{E}^{N*}$. The environmental impact is given as $E^{N*}=e_{n}D_{n}^{N*}+e_{r}D_{r}^{N*}$.

### 4.3. Strategy R (Model R)

In model R, E provides TOR with EWS to consumers. M wholesales new and remanufactured products to E, and then, E sells these products to consumers by two ways: TON and TOR. M and E play a dynamic Stackelberg game. M firstly determines the wholesale prices for new and remanufactured products, and then, E determines the retail prices of new and remanufactured products and the retail price of EWS. The optimization models of profit maximization for M and E are formulated as
(3)$$\begin{array}{l} \underset{w_{n}^{R},w_{r}^{R}}{\max}\pi_{M}^{R}=(w_{n}^{R}-c_{n})D_{n}^{R}+(w_{r}^{R}-c_{r})D_{r}^{R}-A(D_{n}^{R}+D_{r}^{R})\\ s.t.\underset{p_{n}^{R},p_{r}^{R},p_{e}^{R}}{\max}\pi_{E}^{R}=(p_{n}^{R}-w_{n}^{R})D_{n}^{R}+(p_{r}^{R}-w_{r}^{R})D_{r}^{R}+(p_{e}^{R}-t^{2})D_{e}^{R} \end{array}$$

By using reverse induction, we obtain the optimal results for model R, which are described in the following Theorem 3.

**Theorem** **3.***In model R, the optimal prices are given as*$w_{n}^{R*}=\frac{\phi_{1}}{2}+\frac{\phi_{3}}{2(1-b^{2})}$, $w_{r}^{R*}=\frac{\phi_{2}}{2}+\frac{\phi_{4}}{2(1-b^{2})}+\frac{t(k-t)}{4}$, $p_{n}^{R*}=\frac{\phi_{1}}{4}+\frac{3\phi_{3}}{4(1-b^{2})}$, $p_{r}^{R*}=\frac{\phi_{2}}{3}-\frac{b\phi_{1}}{12}+\frac{b\phi_{3}}{12(1-b^{2})}+\frac{2\phi_{4}}{3(1-b^{2})}+\frac{t(t-k)}{6}$*and*$p_{e}^{R*}=\frac{b\phi_{1}}{6}-\frac{\phi_{2}}{6}+\frac{2(1-a)+(5t+7k)t}{12}$.

**Proof.** See Appendix C. 

Thus, the optimal demands for new and remanufactured products are shown as $D_{n}^{R*}=\frac{b\phi_{2}}{12}-\frac{(b^{2}+3)\phi_{1}}{12}+\frac{3a-b+ab+2bt(t-k)}{12}$, $D_{r}^{R*}=\frac{b\phi_{1}}{3}-\frac{\phi_{2}}{3}+\frac{2(1-a)+t(k-t)}{6}$, $D^{R*}=\frac{(1-b)(b-3)\phi_{1}}{12}-\frac{(1-b)\phi_{2}}{3}-\frac{\phi_{3}}{12}+\frac{t(1-b)(k-t)+2}{6}$ and $D_{e}^{R*}=\frac{b\phi_{1}}{6}-\frac{\phi_{2}}{6}+\frac{\phi_{3}}{6}+\frac{7t(k-t)-2ab}{12}$.

The optimal profits of M, E and RSC are shown as $\pi_{M}^{R*}=(w_{n}^{R*}-c_{n})D_{n}^{R*}+(w_{r}^{R*}-c_{r})D_{r}^{R*}-A(D_{n}^{R*}+D_{r}^{R*})$, $\pi_{E}^{R*}=(p_{n}^{R*}-w_{n}^{R*})D_{n}^{R*}+(p_{r}^{R*}-w_{r}^{R*})D_{r}^{R*}+(p_{e}^{R*}-t^{2})D_{e}^{R*}$ and $\pi_{T}^{R*}=\pi_{M}^{R*}+\pi_{E}^{R*}$. The environment impact is given as $E^{R*}=e_{n}D_{n}^{R*}+e_{r}D_{r}^{R*}$.

### 4.4. Strategy NR (Model NR)

In model NR, M provides TON and TOR with EWS to consumers. M wholesales new and remanufactured products to E, and then, E sells these products to consumers by two ways: TON and TOR. M and E play a dynamic Stackelberg game. M firstly determines the wholesale prices for new and remanufactured products, and then, E determines the retail prices of new and remanufactured products and the retail price of EWS. The optimization models of profit maximization for M and E are formulated as
(4)$$\begin{array}{l} \underset{w_{n}^{NR},w_{r}^{NR}}{\max}\pi_{M}^{NR}=(w_{n}^{NR}-c_{n})D_{n}^{NR}+(w_{r}^{NR}-c_{r})D_{r}^{NR}-AD^{NR}\\ s.t.\underset{p_{n}^{NR},p_{r}^{NR},p_{e}^{NR}}{\max}\pi_{E}^{NR}=(p_{n}^{NR}-w_{n}^{NR})D_{n}^{NR}+(p_{r}^{NR}-w_{r}^{NR})D_{r}^{NR}+(p_{e}^{NR}-t^{2})D_{e}^{NR} \end{array}$$

By using reverse induction, we obtain the optimal results for model TR, which are described in the following Theorem 4.

**Theorem** **4.***In model NR, the optimal prices are given as*$w_{n}^{NR*}=\frac{\phi_{1}}{2}+\frac{\phi_{{}_{3}}}{2(1-b^{2})}+\frac{t(k-t)}{4}$, $w_{r}^{NR*}=\frac{\phi_{2}}{2}+\frac{\phi_{4}}{2(1-b^{2})}+\frac{t(k-t)}{4}$, $p_{n}^{NR*}=\frac{(b+3)\phi_{1}}{8(1+b)}+\frac{(1-b)\phi_{2}}{8(1+b)}+\frac{3\phi_{3}}{4(1-b^{2})}+\frac{2t(t-k)-1}{8(1+b)}$, $p_{r}^{NR*}=\frac{(1-b)\phi_{1}}{8(1+b)}+\frac{(b+3)\phi_{2}}{8(1+b)}+\frac{3\phi_{4}}{4(1-b^{2})}-\frac{2t^{2}-2kt+1}{8(1+b)}$*and*$p_{e}^{NR*}=\frac{(b-1)\phi_{1}}{4(b+1)}+\frac{(b-1)\phi_{2}}{4(b+1)}+\frac{(1+3b)t^{2}+(b+3)kt+1}{4(b+1)}$.

**Proof.** See Appendix D. 

Thus, the optimal demands for the new and remanufactured products are shown as $D_{n}^{NR*}=\frac{(b^{2}+4b-1)\phi_{2}}{8(1+b)}-\frac{(3+b^{2})\phi_{1}}{8(1+b)}+\frac{\phi_{4}}{8(1+b)}+\frac{3a+ab-b+2t(b-1)(t-k)}{8(1+b)}$, $D_{r}^{NR*}=\frac{(b^{2}+4b-1)\phi_{1}}{8(1+b)}-\frac{(b^{2}+3)\phi_{2}}{8(1+b)}+\frac{\phi_{3}}{8(1+b)}+\frac{3\phi_{4}}{8(1+b)}+\frac{t(1-b)(k-t)-2ab}{4(1+b)}$, $D^{NR*}=\frac{(b-1)\phi_{1}}{2(b+1)}+\frac{(b-1)\phi_{2}}{2(b+1)}+\frac{t(1-b)(k-t)+1}{2(b+1)}$ and $D_{e}^{NR*}=\frac{(b-1)\phi_{1}}{4(b+1)}+\frac{(b-1)\phi_{2}}{4(b+1)}+\frac{t(k-t)(3+b)+1}{4(b+1)}$.

The optimal profits of M, E and RSC are shown as $\pi_{M}^{NR*}=(w_{n}^{NR*}-c_{n})D_{n}^{NR*}+(w_{r}^{NR*}-c_{r})D_{r}^{NR*}-A(D_{n}^{NR*}+D_{r}^{NR*})$, $\pi_{E}^{NR*}=(p_{n}^{NR*}-w_{n}^{NR*})D_{n}^{NR8}+(p_{r}^{NR*}-w_{r}^{NR*})D_{r}^{NR*}+(p_{e}^{NR*}-t^{2})D_{e}^{NR*}$ and $\pi_{T}^{NR*}=\pi_{M}^{NR*}+\pi_{E}^{NR*}$. The environment impact is given as $E^{NR*}=e_{n}D_{n}^{NR*}+e_{r}D_{r}^{NR*}$.

## 5. Sensitive and Comparative Analysis of Equilibrium Results

By analyzing and comparing the equilibrium results in Theorems 1–4, the following conclusions can be drawn.

By analyzing the effects of the parameters k and t on the optimal wholesale prices in the models N, R and NR, we have the following Proposition 1.

**Proposition** **1.**
*The optimal wholesale prices in the three models satisfy:*


*(1)*$\frac{\partial w_{n}^{N*}}{\partial k}=\frac{\partial w_{n}^{NR*}}{\partial k}=\frac{\partial w_{r}^{R*}}{\partial k}=\frac{\partial w_{r}^{NR*}}{\partial k}>\frac{\partial w_{n}^{R*}}{\partial k}=\frac{\partial w_{r}^{N*}}{\partial k}=0$;

*(2)*$\frac{\partial w_{n}^{N*}}{\partial t}=\frac{\partial w_{n}^{NR*}}{\partial t}=\frac{\partial w_{r}^{R*}}{\partial t}=\frac{\partial w_{r}^{NR*}}{\partial t}$, $\frac{\partial w_{n}^{R*}}{\partial t}=\frac{\partial w_{r}^{N*}}{\partial t}=0$.

**Proof.** See Appendix E. 

Proposition 1(1) indicates that the wholesale prices of new products in the models N and NR, and the wholesale prices of remanufactured products in the models R and NR increase with the increase of consumers’ sensitivity to the length of EWS, and the sensitivities are the same, while the wholesale prices of new products in model R and the wholesale prices of remanufactured products in model N do not change with the change of consumers’ sensitivity to the length of EWS. Proposition 1(2) shows that the effects of the length of EWS on the wholesale prices of new products in the models N and NR are the same as that on the wholesale prices of remanufactured products in the models R and NR. The length of EWS does not affect the wholesale prices of new and remanufactured products in models R and N, respectively.

By comparing the optimal wholesale prices in the models B, N, R and NR, we have the following Proposition 2.

**Proposition** **2.***The optimal wholesale prices in the four models satisfy:*$w_{n}^{N*}=w_{n}^{NR*}$, $w_{n}^{R*}=w_{n}^{B*}$, $w_{r}^{NR*}=w_{r}^{R*}$*and*$w_{r}^{N*}=w_{r}^{B*}$.

**Proof.** See Appendix F. 

Proposition 2 indicates that the wholesale prices of new products in the models N and NR are the same, and they are also the same in the models B and R, while the wholesale prices of the remanufactured products are the same in the models R and NR, and they are also the same in the models B and N. That is to say, M will set the same wholesale prices of new products under strategies N and NR or strategies R and B. M will also set the same wholesale prices of remanufactured products under strategies NR and R or strategies N and B.

By analyzing the effects of the parameters k and t on the optimal retail prices in the models N, R and NR, we have the following Proposition 3.

**Proposition** **3.**
*The optimal retail prices in the three models satisfy:*


*(1)*$\frac{\partial p_{r}^{R*}}{\partial k}<\frac{\partial p_{n}^{R*}}{\partial k}=\frac{\partial p_{r}^{N*}}{\partial k}=0<\frac{\partial p_{n}^{N*}}{\partial k}$, $\frac{\partial p_{r}^{NR*}}{\partial k}=\frac{\partial p_{n}^{NR*}}{\partial k}<0$, $\frac{\partial p_{e}^{N*}}{\partial k}=\frac{\partial p_{e}^{R*}}{\partial k}>0$*and*$\frac{\partial p_{e}^{NR*}}{\partial k}>0$;

*(2)*$\frac{\partial p_{n}^{N*}}{\partial t}=\frac{\partial p_{r}^{R*}}{\partial t}$, $\frac{\partial p_{n}^{R*}}{\partial t}=\frac{\partial p_{r}^{N*}}{\partial t}=0$, $\frac{\partial p_{n}^{NR*}}{\partial t}=\frac{\partial p_{r}^{NR*}}{\partial t}$, $\frac{\partial p_{e}^{N*}}{\partial t}=\frac{\partial p_{e}^{R*}}{\partial t}>0$*and*$\frac{\partial p_{e}^{NR*}}{\partial t}>0$.

**Proof.** See Appendix G. 

Proposition 3(1) indicates that the retail price of new products decreases with the increase of consumers’ sensitivity to the length of EWS in model NR, while this increases with the increase of consumers’ sensitivity to the length of EWS in model N, and the retail price of new products does not change with the increase of parameter k. The retail price of remanufactured products decreases with the increase of consumers’ sensitivity to EWS length k in the models R and NR, while this does not change with the increase of consumers’ sensitivity to the length of EWS in model N. In addition, the retail prices of EWS increases with the increase of consumers’ sensitivity to EWS length k in the models N, R and NR. With the increase of k, E will increase the price of EWS to obtain more market demands for EWS. Proposition 3(2) shows that the effects of EWS’s length on the retail prices of new and remanufactured products in models N and NR are the same, while the retail prices of new products do not change with the increase of EWS’s length t in model R. The retail prices of new products do not change with the increase of EWS’s length t in model N. The retail prices of EWS increase with the increase of EWS’s length t in the models N, R and NR. With the increase of t, E will increase the price of EWS to make up for the loss by the rising EWS cost.

By comparing the optimal retail prices in the models B, N, R and NR, we have the following Proposition 4.

**Proposition** **4.**
*The optimal retail prices in the four models satisfy:*


*(1)*$p_{n}^{B*}=p_{n}^{R*}$, *if*$k>\Phi_{1}$, *then*$p_{n}^{B*}>p_{n}^{N*}$*and*$p_{n}^{R*}>p_{n}^{N*}$; *if*$k>\Phi_{2}$, *then*$p_{n}^{B*}>p_{n}^{NR*}$, *and*$p_{n}^{R*}>p_{n}^{NR*}$;

*(2)*$p_{r}^{B*}=p_{r}^{N*}$; *if*$k>\Phi_{3}$, *then*$p_{r}^{B*}>p_{r}^{R*}$*and*$p_{r}^{N*}>p_{r}^{R*}$; *if*$k>\Phi_{4}$, *then*$p_{r}^{B*}>p_{r}^{NR*}$*and*$p_{r}^{N*}>p_{r}^{NR*}$;

*(3) if*$\Delta >\frac{2a-1}{1+b}$, *then*$p_{e}^{R^{*}}>p_{e}^{N^{*}}$;

*where*$\Phi_{1}=\frac{(\phi_{1}+\phi_{2})(1-b)+2t^{2}-a}{2t}$, $\Phi_{2}=\frac{(\phi_{1}+\phi_{2})(1-b)+2t^{2}-1}{2t}$, $\Phi_{3}=\frac{\phi_{2}-b\phi_{1}+2t^{2}-1}{2t}$*and*$\Phi_{4}=\frac{(\phi_{1}+\phi_{2})(1+b)+2t^{2}-1}{2t}$.

**Proof.** See Appendix H. 

Proposition 4(1) indicates that the retail prices of new products are the same in the models B and R, and the retail prices of remanufactured prices of remanufactured products are the same in the models B and N. In addition, when consumers’ sensitivity to the length of EWS is more than a certain value, the retail price of new products in model B and the retail price of new products in model R are higher than these in model NR, respectively. When consumers’ sensitivity to the length of EWS is more than a certain value, the retail price of new products in model B is more than that in model N, and the retail price of new products in model N is lower than that in model R. From Proposition 4(2), we can find that, when consumers’ sensitivity to the length of EWS is more than a certain value, the retail prices of remanufactured products in the models B and N are all higher than these in model R. The retail prices of remanufactured products in the models B and N are more than these in model NR. That is to say, products without an EWS strategy or new products with an EWS strategy are better than remanufactured products with EWS. Thus, E will not choose remanufactured products with EWS in order to guarantee the retail prices of remanufactured products. Proposition 4(3) shows that, when the cost-saving Δ is more than a certain value, E will set higher retail prices of the EWS in model R than that in model N.

By analyzing the effects of parameters k and t on the optimal demands in the models N, R and NR, we have the following Proposition 5.

**Proposition** **5.**
*The optimal demands in the three models satisfy:*


*(1)*$\frac{\partial D_{n}^{N*}}{\partial k}=\frac{\partial D_{r}^{R*}}{\partial k}>\frac{\partial D_{n}^{R*}}{\partial k}=\frac{\partial D_{r}^{N*}}{\partial k}$, $\frac{\partial D_{n}^{NR*}}{\partial k}=\frac{\partial D_{r}^{NR*}}{\partial k}>0$, $\frac{\partial D_{e}^{N*}}{\partial k}=\frac{\partial D_{e}^{R*}}{\partial k}>0$*and*$\frac{\partial D_{e}^{NR*}}{\partial k}>0$;

*(2)*$\frac{\partial D_{n}^{N*}}{\partial t}=\frac{\partial D_{r}^{R*}}{\partial t}$, $\frac{\partial D_{n}^{R*}}{\partial t}=\frac{\partial D_{r}^{N*}}{\partial t}$, $\frac{\partial D_{n}^{NR*}}{\partial t}=\frac{\partial D_{r}^{NR*}}{\partial t}$*and*$\frac{\partial D_{e}^{N*}}{\partial t}=\frac{\partial D_{e}^{R*}}{\partial t}$.

**Proof.** See Appendix I. 

Proposition 5(1) indicates that the demands of new products increase with the increase of consumers’ sensitivity to the length of EWS in the models N and NR, while the demand for new products decreases with the increase of consumers’ sensitivity to the length of EWS in the model R. The demands of remanufactured products are increasing with the increase of consumers’ sensitivity to the length of EWS in the models R and NR, while that is decreasing with the increase of consumers’ sensitivity to the length of EWS in the model N. The demands of EWS are increasing with the increase of consumers’ sensitivity to the length of EWS in the models N, R and NR. Proposition 5(2) shows that the effects of EWS’s length on the demands of new and remanufactured products in the models N and R, or models R and N are the same, respectively. The effects of EWS’s length on the demands of new and remanufactured products in the model NR are the same. Moreover, the effects of EWS’s length on the demand of EWS in models N and R are the same, respectively. It can be suggested that RSC members should affect the consumers and provide a suitable EWS’s length to explore more demand for remanufactured products.

By comparing the optimal demands in the models B, N, R and NR, we have the following Proposition 6.

**Proposition** **6.**
*The optimal demands in the four models satisfy:*


*(1) if*$k<\Phi_{5}$, *then*$D_{n}^{B*}>D_{n}^{N*}$, *if*$k>\Phi_{6}$, *then*$D_{n}^{B*}>D_{n}^{R*}$, *if*$k<\Phi_{2}$, *then*$D_{n}^{B*}>D_{n}^{NR*}$;

*(2) if*$k>\Phi_{7}$, *then*$D_{r}^{B*}>D_{r}^{N*}$; *if*$k<\Phi_{6}$, *then*$D_{r}^{B*}>D_{r}^{R*}$; *if*$k<\Phi_{2}$, *then*$D_{r}^{B*}>D_{r}^{NR*}$;

*(3) if*$\Delta >\frac{2a-1}{1+b}$, *then*$D_{e}^{R^{*}}>D_{e}^{N^{*}}$;

*where,*$\Phi_{5}=\frac{\phi_{1}-b\phi_{2}+2t^{2}+a}{2t}$, $\Phi_{6}=\frac{\phi_{2}-b\phi_{1}+2t^{2}+a-1}{2t}$*and*$\Phi_{7}=\frac{\phi_{1}-b\phi_{2}+2t^{2}-a}{2t}$.

**Proof.** See Appendix J. 

Proposition 6(1) indicates that, when consumers’ sensitivity to the length of EWS is lower than a certain value, the demand for new products in model N is lower than that in model B, and the demand for new products in model B is higher than that in model R. When the consumers’ sensitivity to the length of EWS satisfies some conditions, the demand for new products in model B is higher than that in model NR. Proposition 6(2) shows that, differently from the demand for new products, when consumers’ sensitivity to the length of EWS is more than a certain value, the demand for remanufactured products in model B is higher than that in model N. When consumers’ sensitivity to the length of EWS is lower than a certain value, the demand for remanufactured products in model B is higher than that in the models R and NR. It is suggested that RSC members can affect consumers, increasing the recycling of used products, by using the EWS strategy. Proposition 6(3) shows that, when the cost-saving Δ is more than a certain value, the demand of EWS in model R is higher than that in model N.

By analyzing the effects of the parameters k and t on the optimal profits in the models N, R and NR, we have the following Proposition 7.

**Proposition** **7.**
*The optimal profits in the three models satisfy:*


*(1) if*$k>\Phi_{8}$, *then*$\frac{\partial \pi_{M}^{N*}}{\partial k}>0$; *if*$k>\Phi_{9}$, *then*$\frac{\partial \pi_{M}^{R*}}{\partial k}>0$; *if*$k>\Phi_{10}$, *then*$\frac{\partial \pi_{M}^{NR*}}{\partial k}>0$; *if*$k>\Phi_{11}$, *then*$\frac{\partial \pi_{E}^{N*}}{\partial k}>0$, *if*$k>\Phi_{12}$, *then*$\frac{\partial \pi_{E}^{R*}}{\partial k}>0$; *if*$k>\Phi_{13}$, *then*$\frac{\partial \pi_{E}^{NR*}}{\partial k}>0$.

*(2) if*$k>\max\{2t,\Phi_{8}\}$*or*$k<\min\{2t,\Phi_{8}\}$, *then*$\frac{\partial \pi_{M}^{N*}}{\partial t}>0$; *if*$k>\max\{2t,\Phi_{9}\}$*or*$k<\min\{2t,\Phi_{9}\}$, *then*$\frac{\partial \pi_{M}^{R*}}{\partial t}>0$; *if*$k>\max\{2t,\Phi_{11}\}$*or*$k<\min\{2t,\Phi_{11}\}$, *then*$\frac{\partial \pi_{E}^{N*}}{\partial t}>0$; *if*$k>\max\{2t,\Phi_{10}\}$*or*$k<\min\{2t,\Phi_{10}\}$, *then*$\frac{\partial \pi_{M}^{NR*}}{\partial t}>0$; *if*$k>\max\{2t,\Phi_{12}\}$*or*$k<\min\{2t,\Phi_{12}\}$, *then*$\frac{\partial \pi_{E}^{R*}}{\partial t}>0$; *if*$k>\max\{2t,\Phi_{14}\}$*or*$k<\min\{2t,\Phi_{14}\}$, *then*$\frac{\partial \pi_{E}^{NR*}}{\partial t}>0$,

*where,*$\Phi_{8}=\frac{2(\phi_{1}-b\phi_{2})+t^{2}-2a}{t}$, $\Phi_{9}=\frac{2(\phi_{2}-b\phi_{1})-2+2a+t^{2}}{t}$, $\Phi_{10}=\frac{(\phi_{1}+\phi_{2})(1-b)+t^{2}(1-b)-1}{t(1-b)}$, $\Phi_{11}=\frac{2(\phi_{1}-b\phi_{2})+13t^{2}-2a}{13t}$, $\Phi_{12}=\frac{2(\phi_{2}-b\phi_{1})+13t^{2}+2a-2}{13t}$, $\Phi_{13}=\frac{(\phi_{1}+\phi_{2})(1-b)-3bt^{2}-t^{2}-1}{t(5+3b)}$*and*$\Phi_{14}=\frac{(\phi_{1}+\phi_{2})(1-b)+5t^{2}+3bt^{2}-1}{5t(1+3b)}$.

**Proof.** See Appendix K. 

Proposition 7 shows that, when consumers’ sensitivity to the length of EWS meets a certain range, the profits of M and E all increase with the increases of the consumers’ sensitivity k and EWS length t. It can be suggested that manufacturers and E-commerce platforms can cooperate to provide EWS for a longer period of time to improve their profits and operational efficiency of the supply chain system. We can suggest that RSC members should provide an EWS strategy and increase the EWS length to obtain more profits. 

## 6. Numerical Analysis

In order to illustrate the impact of relevant parameters in the supply chain on the decisions of the RSC, a numerical experiment is firstly used in this section. Next, the effects of the parameters on consumer surplus, environment impact and social welfare are also given in this numerical analysis. Following Zhang et al. [11], the parameters are set as $c_{n}=0.4$, $c_{r}=0.3$, t=0.2, A=0.4, a=0.6 and v=0.5. In order to ensure that all the optimal results are positive, k and b vary within the range of [0.5,0.9].

### 6.1. Impacts of Parameters k and b on Decisions of RSC

In this section, the impacts of parameters k and b on the optimal demands and profits are analyzed, and the optimal results are also compared in the four models B, N, R and NR, respectively. The impacts of parameters k and b on the demands of new products are described in Figure 2.

Figure 2 shows that the optimal demands of new products increase with the increase of the parameters k and b in the four models. Figure 2 indicates that, when the level of substitutability between remanufactured and new products is high, E will choose to provide EWS for new products to improve the demand for new products.

The impact of parameters k and b on the demand for remanufactured products are described in Figure 3.

Figure 3 shows that the optimal demand for remanufactured products increases with the increase of the parameters k and b in the four models. In addition, the demand for remanufactured products in model R is the highest. E will choose to provide EWS for remanufactured products to improve the demand for remanufactured products. The demand for remanufactured products increases under the strategy: remanufactured products with EWS, which suggests that an extended warranty service is beneficial to the sale of remanufactured products. In practice, consumers are always more willing to buy products with EWS, which is good for recycling and environment protection.

The impact of parameters k and b on the total demands is described in Figure 4.

Figure 4 shows that the total demand increases with the increase of the parameters k and b in the four models. In addition, the total demand in model NR is the highest. From the perspective of increasing the total demand, E will choose to provide EWS for both new products and remanufactured products.

The impact of parameters k and b on the demands of EWS is described in Figure 5.

Figure 5 indicates that the demands of EWS in the models N, R and NR increase with the increase of the parameters k and b. Figure 5 shows that the demand of EWS in model NR is the highest in the three models. E will choose to provide EWS for new and remanufactured products to improve the demand of EWS. Only consumers who have purchased new products or remanufactured products will choose to purchase EWS, and the total demand of the products in model NR is the highest. Therefore, E will provide EWS to new and remanufactured products to improve the demand of EWS.

The impact of parameters k and b on the optimal profits of M s described in Figure 6.

Figure 6a shows that, in the case of the above values, the profits of M increase with the increase of the parameters k and b in the four models. Figure 6b shows that the profit of M in model NR is the highest. From the above propositions, the wholesale prices of new and remanufactured products increase with the increase of the parameters k and b in model NR, and the demand of the total products in model NR is the highest, so, in order to maximize its own profit, M will take measures to allow E to provide EWS to new and remanufactured products.

The impact of parameters k and b on the optimal profits of E is described in Figure 7.

Figure 7a indicates that, in the case of the above values, the profits of E in the four models all increase with the increase of the parameters k and b. Figure 7b shows that the profit of E in model NR is the highest. From the above propositions, the retail prices of new and remanufactured products increase with the increase of the level of substitutability between remanufactured and new products, while these decrease with the increase of parameter k. The demand of the total products increases with the increase of the parameters k and b, the demand of the total products in model NR is far greater than in the other models. In order to maximize its own profit, E will choose to provide EWS to new and remanufactured products.

The impact of parameters k and b on the optimal total profits is described in Figure 8.

Figure 8a shows that the changing trend of the profit of CLSC is the same as the changing trend of most supply chain members’ profits; that is to say, the profit of CLSC in the four models all increase with the increase of the parameters k and b in the case of the above values. Figure 8b shows that the profit of CLSC in model NR is the highest. From the perspective of CLSC, the supply chain will choose new and remanufactured products with this service strategy. This strategic choice is the same as that of M and E.

### 6.2. Effects of Parameters k and b on Consumer, Environment and Society

The impacts of parameters k and b the consumer surplus, environment impact and social welfare are analyzed in this section.

Following Xiang and Xu [76], the consumer surplus for new and remanufactured products in an RSC can be calculated as $CS=D^{2}/2$. Thus, the impact of k and b on consumer surplus is shown in the following Figure 9.

Figure 9 indicates that consumer surpluses increase with the increase of the parameters k and b in the models R, N and B, while this decreases with the increase of the level of substitutability between remanufactured and new products. According to the calculation formula of consumer surplus, it can be seen that product demand affects consumer surplus. From the above propositions, the demand of the total products in model NR is the highest, so the consumer surplus in model NR is the highest. From the perspective of consumers, the new and remanufactured products with this service strategy is better.

Following Zhang et al. [11], Ding et al. [75], Liu et al. [17] and Yenipazarli et al. [77], the environmental impact of new and remanufactured products can be calculated as $E=e_{n}\times D_{n}+e_{r}\times D_{r}$. Since $e_{n}>0$ and $e_{r}>0$, we set $e_{n}=0.3$ and $e_{r}=0.1$ in this section. Thus, the effects of k and b on the environment are shown in the following Figure 10.

Figure 10 indicates that, with the increase of k and b, the environmental impacts increase in the models B, N, R and NR. When k and b are small, the environmental impact in model R is the lowest, and the environmental impact in model NR is the highest. When k and b are high, the environmental impact in model R is the lowest, while the environmental impact in model N is the highest. We can also find that remanufactured products with an EWS strategy can decrease the environmental impact, and from the perspective of environmental protection, E should adopt strategy R, which also means offering extended warranties to remanufactured products, while that will harm the benefits of supply chain members and consumers.

Following Zhang et al. [11] and Geng and Fan [78], social welfare is composed of the profits of supply chain members, consumer surplus and environmental impact costs. Therefore, social welfare can be calculated as $SW=\pi_{M}+\pi_{E}+CS-vE$. The effects of k and b on the environment are shown in the following Figure 11.

Figure 11a indicates that social welfare always increases with the increase of consumers’ EWS time sensitivity and product cross-price elasticity coefficient. Figure 11b shows that, when E provides EWS to new and remanufactured products, social welfare is the highest. When k and b are small, the change in social welfare is small, and when k and b are high, the change in social welfare is high.

Figure 6, Figure 7, Figure 8, Figure 9, Figure 10 and Figure 11 show that M’s profit, E’s profit and the social welfare trends are roughly the same. It is always better for E to provide EWS than to not provide EWS. New and remanufactured products with an EWS strategy are the most beneficial to the supply chain, consumers and society; this is a strategy jointly decided by the supply chain, consumers and government.

## 7. Conclusions

This study’s main object was to put forward an optimal extended warranty strategy for a remanufactured supply chain under “trade old for new” and “trade old for remanufactured”. Due to the aggravation of environmental pollution, China’s State Council resource administration has issued regulations on the recycling and disposal of waste electrical and electronic products and adjusted the subsidy standard of the waste electrical and electronic products treatment fund (http://www.gov.cn/zhengce/zhengceku/2021-03/23/content_5595129.htm, accessed on 22 March 2021). Specifically, the extended warranty strategy with “trade old for new” and “trade old for remanufactured” is one way of reducing carbon emissions, which has been used in many countries, such as the USA, China and Korea. The regulations that exist in the field of “trade old for new”, “trade old for remanufactured” and extended warranty strategy have been promulgated (https://baijiahao.baidu.com/s?id=1669897974720139149&wfr=spider&for=pc, accessed on 19 June 2020). Our research points out that the extended warranty service strategy can promote the recycling and remanufacturing of products, which is beneficial to environmental protection. The research results also provide a reference for the extended warranty strategy decision of the E-commerce platform. 

This paper combined new and remanufactured products and EWS, and four models were constructed: B, N, R and NR. The decisions of the RSC members were investigated, and the effects of the EWS on consumer surplus, environment impact and society welfare were analyzed. The results showed that, when the level of substitutability between remanufactured and new products meets a certain range, new products with an EWS (strategy *N*) are more beneficial for the demand for new products, remanufactured products with EWS (strategy *R*) are more beneficial for the demand for remanufactured products and new and remanufactured products with an EWS (strategy *NR*) are beneficial for the total demand for new and remanufactured products. In the four models, the changing trends of manufacturer’s profit, E-commerce platform’s profit and supply chain system’s profit and consumer surplus were the same but only had different change the thresholds of the parameters. Moreover, from the perspective of remanufactured supply chain members, remanufactured supply chain system and society, new and remanufactured products with EWS is the best choice.

Based on the above results, we can draw some management insights. Firstly, extended warranty service affects the demand of new and remanufactured products, and choosing the proper strategy can promote the sale of new and remanufactured products, which can also further promote recycling and remanufacturing. It is beneficial for the protection of the environment. Secondly, the profit for remanufactured supply chain members, consumers and society under the new and remanufactured products with EWS (strategy *NR*) is the highest, and in this case, the supply chain members will actively accept EWS. Thirdly, from the perspective of environmental protection, the impact on the environment will increase under strategy *NR*, and the supply chain will need to take measures to mitigate the environmental impact.

This study reviewed the literature on the theory and practice of extended warranty service, “trade old for new” and “trade old for remanufactured”, which have been implemented in many countries. In October 2021, the CPC Central Committee and the State Council issued their opinions on completely, accurately and comprehensively implementing a new development concept and doing a good job in carbon peaking and carbon neutralization, and the State Council issued the action plan for carbon peaking before 2030. Based on the peaking objective, carbon emissions in the heavy chemical industry should receive the most attention due to its large proportion with respect to the emission sources from other sectors. The proportion of carbon emissions in the energy processing industry, steel industry and building material industry account for a larger fraction of the cumulative carbon emission in the heavy chemical industry during the simulation period in 2035 [79]. Sun et al. [80] found that China can potentially reduce the CO_2_ emission intensity (CEI) by up to 72.7% compared to the 2005 level in 2030 and, therefore, that the official CEI reduction targets (60–65%) are well within reach. The CEI will not readily converge in future years due to the large differences in the energy-saving and emission abatement potential across different regions in China; the rank of convergence is CO_2_ > CO_2_ per capita > CEI. Hebei, Shandong, Shanxi, Liaoning, Inner Mongolia and Xinjiang are key areas in whether or not their respective regions complete the emission reduction task. This also confirms the significance of our research.

There were still some limitations in this paper. First of all, in this paper, we considered the EWS length as a parameter. In other words, we did not consider how the warranty service length can influence the prices, the demands and the profits. In the future, we can consider the service length as a decision variable and investigate the influence of the service length. Secondly, in this paper, the cost of the EWS was offered by the E-commerce platform. In the future, we can improve the management of the supply chain by considering cooperation between manufacturers and the E-commerce platform and considering cost-sharing. Finally, our paper only considered a manufacturer and an E-commerce platform. In the future, we can consider the competition between two or more manufacturers or competition between two or more E-commerce platforms.

## Figures and Tables

**Figure 1 ijerph-19-01526-f001:**
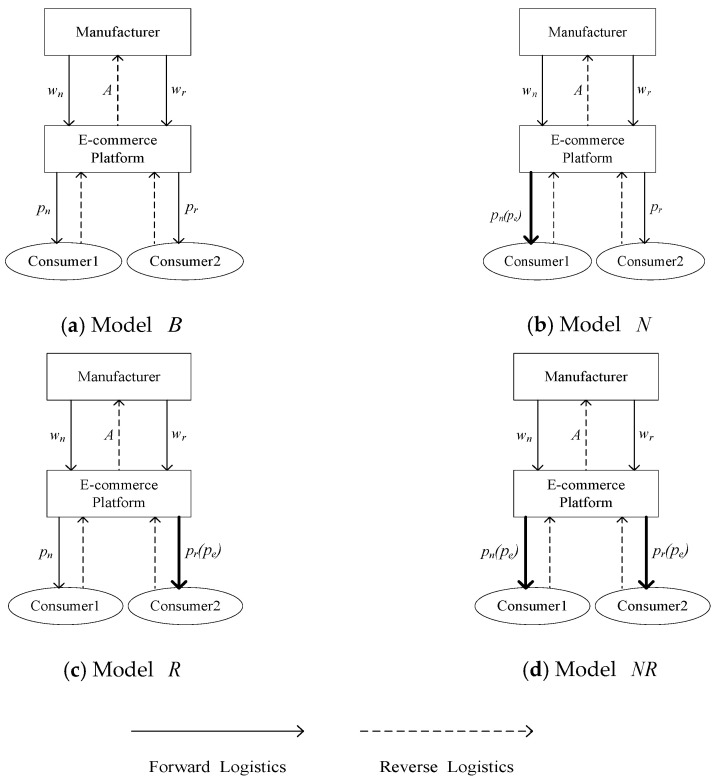
Structure of the four models in a CLSC.

**Figure 2 ijerph-19-01526-f002:**
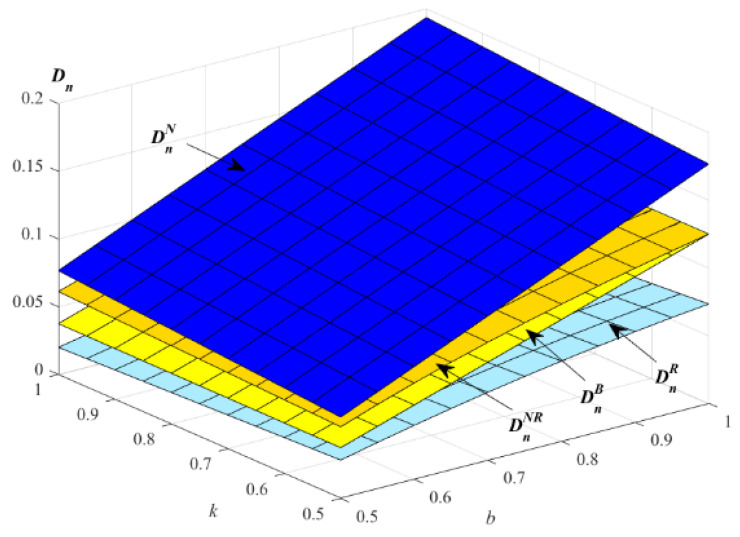
Impact of *k* and *b* on the optimal demand of new products.

**Figure 3 ijerph-19-01526-f003:**
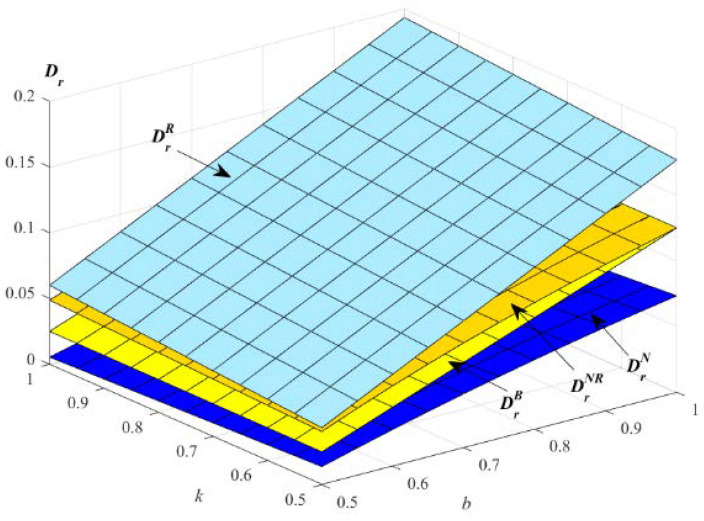
Impact of *k* and *b* on the optimal demand for remanufactured products.

**Figure 4 ijerph-19-01526-f004:**
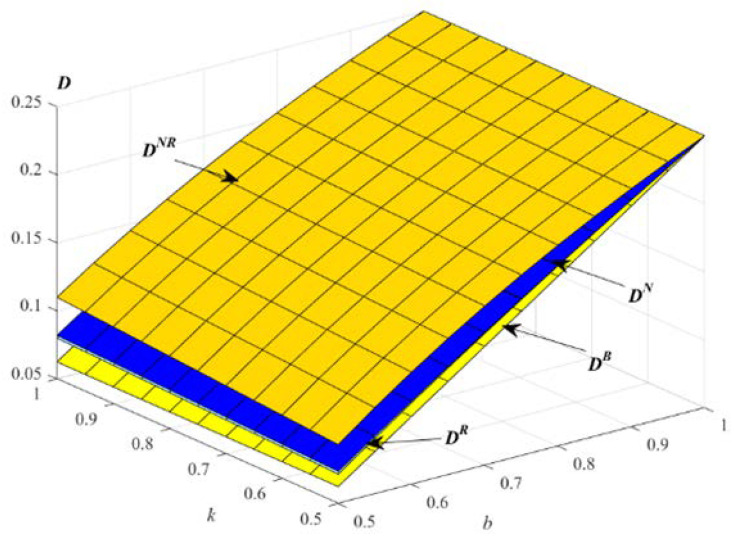
Impact of *k* and *b* on the optimal total demand.

**Figure 5 ijerph-19-01526-f005:**
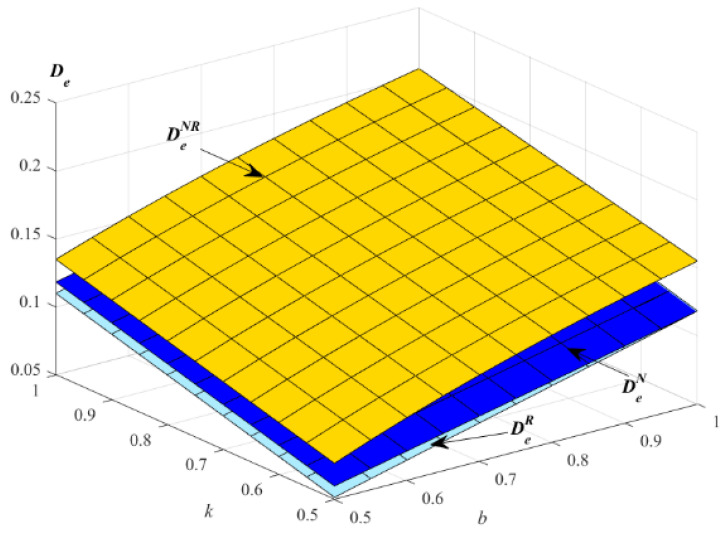
Impact of *k* and *b* on the optimal demand of EWS.

**Figure 6 ijerph-19-01526-f006:**
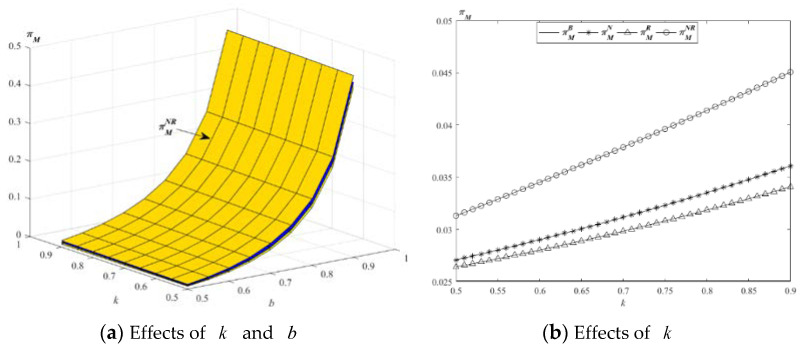
Impact of *k* and *b* on the optimal profit of M.

**Figure 7 ijerph-19-01526-f007:**
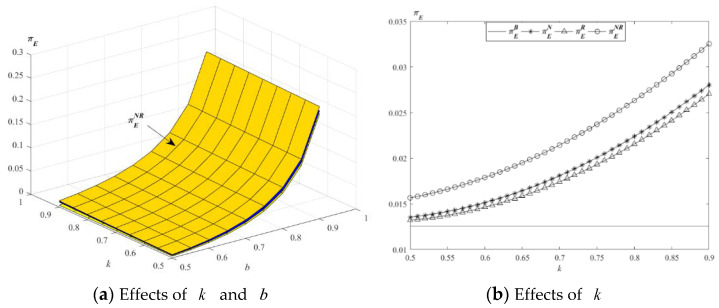
Impact of *k* and *b* on the optimal profit of E.

**Figure 8 ijerph-19-01526-f008:**
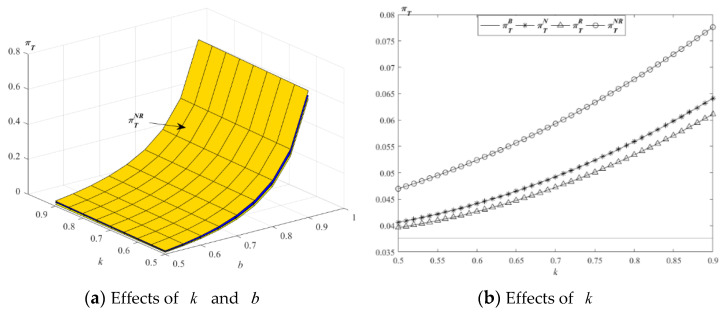
Impact of *k* and *b* on the optimal profit of CLSC.

**Figure 9 ijerph-19-01526-f009:**
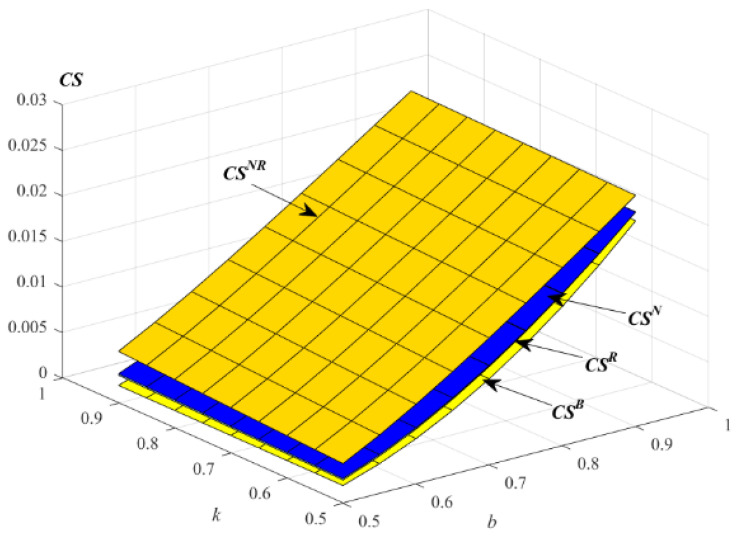
Impact of *k* and *b* on the consumer surplus.

**Figure 10 ijerph-19-01526-f010:**
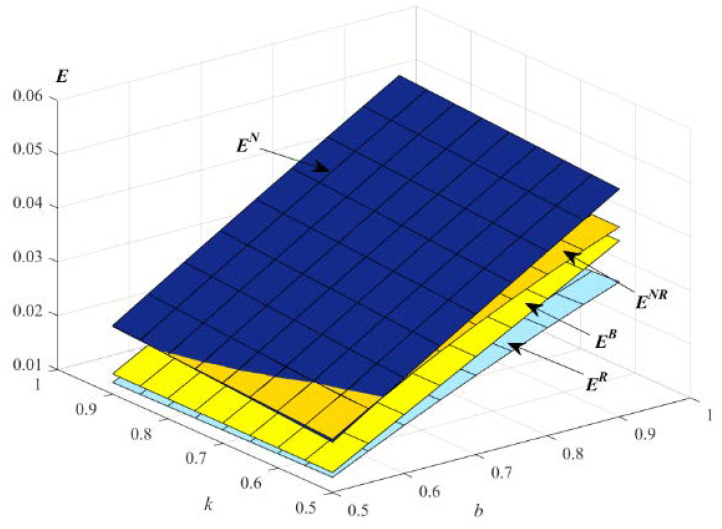
Impact of *k* and *b* on the environment.

**Figure 11 ijerph-19-01526-f011:**
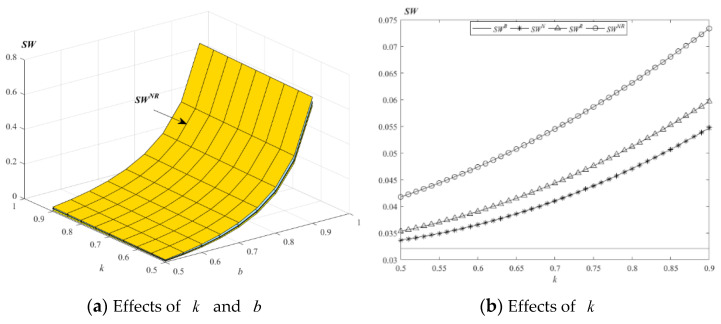
Changes in social welfare with parameters *k* and *b*.

**Table 1 ijerph-19-01526-t001:** Parameters and decision variables.

Notations	Definition
a	Proportion of new product market
b	Cross-price elasticity coefficient
t	Length of EWS
k	Consumers’ sensitivity to length of EWS
$c_{n}/c_{r}$	Unit production cost of new/remanufactured products
A	Unit price of recycling used products
v	Environmental impact cost coefficient
$e_{n}/e_{r}$	Unit environmental impact of new/remanufactured products
$w_{n}/w_{r}$	Unit wholesale price of new products/remanufactured products
$p_{n}/p_{r}$	Unit retail price of new/remanufactured products
$p_{e}$	Unit price of the EWS
$D_{e}$	Demand of the EWS products
$D_{n}^{j}/D_{r}^{j}/D^{j}$	Demand for new/remanufactured/total products in model j , j∈{B,N,R,NR}
$\pi_{i}^{j}$	Profit of supply chain member i in model j, i∈{M,E,T}, j∈{B,N,R,NR}
$E^{j}$	Environmental impact in model j , j∈{B,N,R,NR}
$CS^{j}$	Consumer surplus in model j , j∈{B,N,R,NR}
$SW^{j}$	Social welfare in model j , j∈{B,N,R,NR}

## Data Availability

The data came from the literature and management practices, and all the data had literature sources.

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
