# Peer review of "Extended Warranty Strategy and Its Environment Impact of Remanufactured Supply Chain"

_ijerph, 2022, doi:10.3390/ijerph19031526_

Round 1
Reviewer 1 Report
Referee report on
Extended Warranty Strategy and its Environment Impact of Remanufactured Supply Chain
Xuemei Zhang, Jiawei Hu, Suqin Sun and Guohu Qi
The manuscript considering whether the E-commerce platform provides extended warranty service, and four models are constructed and the extended warranty service strategy and its environment impact was analyzed. The results show that: when the level of substitutability between remanufactured and new products meets a certain rage, new or remanufactured products with extended warranty service strategy can increase the demand of new or remanufactured products, respectively. In the four models, the changing trends of manufacturer’s profit, E-commerce platform’s profit and supply chain’s profit, consumer surplus, environmental impact and social welfare are the same, but only the thresholds are different. From the perspective of supply chain member, supply chain system, consumer, environment and society, new and remanufactured products with extended warranty service strategy is the best choice.
Comment:
The paper can be published in its current form since presents interesting and useful content. English is well written, the text of the article is well organized and the notation is easy to follow. The introduction provides sufficient background and includes all relevant references. The research design is appropriate. The method is adequately described. The results are clearly presented, and the conclusions are supported by the results.

Author Response
We are very grateful to the Editors and the three anonymous reviewers for offering constructive comments on our submission and giving us the opportunity to revise this manuscript. We greatly appreciate the two reviewers’ overall endorsement of our research and supportive comments and suggestions to improve the content and presentation of our article. Please find below our point-to-point response to the two reviewers’ comments. Specifically, we first reproduce your comments (in italic) and, then, respond and explain how your comments are addressed in the revised manuscript. All changes are highlighted in yellow in the main text.
RESPONSE TO REVIEWER #1
The paper can be published in its current form since presents interesting and useful content. English is well written; the text of the article is well organized and the notation is easy to follow. The introduction provides sufficient background and includes all relevant references. The research design is appropriate. The method is adequately described. The results are clearly presented, and the conclusions are supported by the results.
Authors’ response: Thank you for your recognition and appreciation of our research.
Reviewer 2 Report
1. Lack of information about climate protection processes, it is necessary to mention solutions related to the reduction of carbon dioxide emissions and decarbonisation,
The processes related to the extended warranty and the closed circulation of the economy have a very large impact on the achievement of environmental protection objectives.
It is necessary to add in the introduction to the article and in the literature research some information on this subject, the world's largest regulations regarding the need to protect the climate.
This should also be added in the summary to this work. So that all these activities are placed in the context of climate protection.
Articles can be used
https://www.tandfonline.com/doi/full/10.1080/14693062.2019.1623164
https://www.mdpi.com/1996-1073/14/5/1217
https://link.springer.com/article/10.1007%2Fs10980-016-0404-8
2. It should be clearly shown, on the basis of two or three examples, the regulations that exist in the field of TON, TOR and EWS in individual US, EU, China markets, whether the rest of Asia, whether and what regulations exist or are planned to be implemented. I do not require it to be very broad, but it should be mentioned in the introduction and at the end in the summary.
3. What is the CSR mentioned in line 103, does this take into account Customer Social Responsibility? How do you envision this contribution? Please add some examples.
4. Do you calculate carbon dioxide emissions hidden in the energy produced and used to produce the product? Do you only take the carbon emissions used in the production of the product itself? Line 195
5. Have there been consumer opinion surveys, do they want products such as TON and TOR or EWS? Add info about it in the literature review.
6. How do these proposals relate to the regulations implemented in the world concerning, for example, returnable bottles and recycling fees? Are there such solutions in China? Add information about it in the summary.
Author Response
We are very grateful to the Editors and the three anonymous reviewers for offering constructive comments on our submission and giving us the opportunity to revise this manuscript. We greatly appreciate the two reviewers’ overall endorsement of our research and supportive comments and suggestions to improve the content and presentation of our article. Please find below our point-to-point response to the two reviewers’ comments. Specifically, we first reproduce your comments (in italic) and, then, respond and explain how your comments are addressed in the revised manuscript. All changes are highlighted in yellow in the main text.
- Lack of information about climate protection processes, it is necessary to mention solutions related to the reduction of carbon dioxide emissions and decarbonization. The processes related to the extended warranty and the closed circulation of the economy have a very large impact on the achievement of environmental protection objectives. It is necessary to add in the introduction to the article and in the literature research some information on this subject, the world's largest regulations regarding the need to protect the climate. This should also be added in the summary to this work. So that all these activities are placed in the context of climate protection.
Articles can be used
https://www.tandfonline.com/doi/full/10.1080/14693062.2019.1623164
https://www.mdpi.com/1996-1073/14/5/1217
https://link.springer.com/article/10.1007%2Fs10980-016-0404-8
Authors’ response and revision: Thanks to your thoughtful suggestion, we have updated the Introduction (Please refer to pages 1, 2), Literature review (Please refer to pages 3, 4) and Conclusion (Please refer to pages 19, 20) sections by incorporating into this revision some recently published papers in solutions related to the reduction of carbon dioxide emissions and regulations. In the whole revised manuscript, we also conduct a more systematic review and additional relevant literature has been added as listed below.
References
- Nash, S.L.; Steurer, R. Climate change acts in Scotland, Austria, Denmark and Sweden: the role of discourse and deliberation. Clim. Policy. 2021, 21(9), 1-12.
- Nash, S.L.; Steurer, R. Taking stock of climate change acts in Europe: living policy processes or symbolic gestures? Clim. Policy. 2019, 19(8), 1052-1065.
- Zhu, X.X.; Ren, M.L.; Wu, G.D.; Pei J.; Pardalos, P.M. Promoting new energy vehicles consumption: the effect of implementing carbon regulation on automobile in China. Comput. Ind. Eng. 2019, 125: 211-226.
- Kristofferse, E.; Blomsma, F.; Mikalef, P.; Li, J.Y. The smart circular economy: a digital-enabled circular strategies framework for manufacturing companies. J. Bus. Res. 2020, 120, 241-261.
- Ye, F.; Ni, D.B.; Li, K.W. Competition between manufacturers and sharing economy platforms: an owner base and sharing utility perspective. Int. J. Prod. Econ. 2021, 234, 108022.
- Liu, C.Y.; Wang, H.; Tang, J.; Chang, C.T.; Liu, Z. Optimal recovery model in a used batteries closed-loop supply chain considering uncertain residual capacity. Transport. Res. E Logist. Transport Rev. 2021, 156, 102516.
- Zheng, Y.Y.; Zhao, Y.X.; Wang, N.G.; Meng, X.G.; Yang, H.L. Financing decision for a remanufacturing supply chain with a capital constrained retailer: a study from the perspective of market uncertainty. Int. J. Prod. Econ. 2022, 245, 108397.
- Zheng, X.X.; Li, D.F.; Liu, Z.; Jia, F.; Lev, B. Willingness-to-cede behavior in sustainable supply chain coordination. Int. J. Prod. Econ. 2021, 240, 108207.
- Nash, S.L.; Steurer, R. From symbolism to substance: what the renewal of the Danish climate change act tells us about the driving forces behind policy change. Environ. Pollut. 2021, doi: 10.1080/096440.2021.1922186.
- Lverson, L.R.; Thompson, F.R.; Matthews, S.; Peters, M.; Prasad, A.; Dijak, W.D.; Fraser, J.; Wang, W.J.; Hanberry, B.; He, H. Multi-model comparison on the effects of climate change on tree species in the eastern U.S.: results from an enhanced niche model and process-based ecosystem and landscape models. Landsc. Ecol. 2017, 32, 1327-1346.
- Wójcik-Jurkiewicz, M.; Czarbecja, M.; Kinelshi, G.; Sadowska, B.; BiliÅ„ska-Reformat, K. Determinants of decarbonization in the transformation of the energy sector: the case of Poland. Energy. 2021, 14, 1217.
- Kristoffersen, E.; Mikalef, P.; Blomsma, F.; Li, J.Y. The effects of business analytics capability on circular economy implementation, resource orchestration capability, and firm performance. Int. J. Prod. Econ. 2021, 239, 108205.
- Liu, B.; Shen, L.J.; Xu, J.Y.; Zhao, X.J. A complimentary extended warranty: profit analysis and pricing strategy. Int. J. Prod. Econ. 2020, 229, 107860.
- Liu, A.J.; Zhang, Y.; Luo, S.H.; Miao, J. Dual-channel global closed-loop supply chain network optimization based on random demand and recovery rate. Int. J. Environ. Res. Public Health. 2020, 17, 8768.
- Kristoffersen, E.; Mikalef, P.; Blomsma, F.; Li, J.Y. Towards a business analytics capability for the circular economy. Technol. Forecast. Soc. Change. 2021, 171, 120957.
- Zhang, A.; Wang, J.X.; Farooque, M.; Wang, Y.L.; Choi, T.M. Multi-dimensional circular supply chain management: a comparative review of the state-of-the-art practices and research. Transp. Res. Part E Logist. Transp. Rev. 2021, 155, 102509.
- Bian, J.S.; Zhao, X. Competitive environmental sourcing strategies in supply chains. Int. J. Prod. Econ. 2020, 230, 107891.
- Ma, L.; Zhang, X.R.; Du, Y.S. Influence mechanism on supplier emission reduction based on a two-level supply chain. Int. J. Environ. Res. Public Health. 2021, 18, 12439.
- Giri, B.C.; Glock, G.H. The bullwhip effect in a manufacturing/remanufacturing supply chain under a price-induced non-standard ARMA (1,1) demand process. Eur. J. Oper. Res. 2021, doi: 10.1016/j.ejor.2021.10.025.
- Luo, R.L.; Zhou, L.; Song, Y.; Fan, T.J. Evaluating the impact of carbon tax policy on manufacturing and remanufacturing decisions in a closed-loop supply chain. Int. J. Prod. Econ. 2022, 245, 108408.
- Zhou, Y.; Xiong, Y.; Jin, M.Y. Less is more: consumer education in a closed-loop supply chain with remanufacturing. Omega. 2021, 101, 102259.
- Yang, F.; Wang, M.M.; Ang, S. Optimal remanufacturing decisions in supply chains considering consumers’’ anticipated regret and power structures. Transp. Res. Part E Logist. Transp. Rev. 2021, 148, 102267.
- Zhang, W.; He, Y. Optimal policies for new and green remanufactured short-life-cycle products considering consumer behavior. J. Clean. Prod. 2019, 214, 483-505.
- Gan, S.S.; Pujawan, N.; Suparno, Widodo, B. Pricing decision for new and remanufactured product in a closed-loop supply chain with separate sales-cahnnel. Int. J. Prod. Econ. 2016, 190, 120-132.
- Hazen, B.; Overstreet, R.E.; Jones-Farmer, L.A.; Field, H.S. The role of ambiguity tolerance in consumer perception of remanufactured products. Int. J. Prod. Econ, 2012, 135(2): 781-790.
- Han, Y.M.; Li, J.Z.; Lou, X.Y.; Fan, C.Y.; Geng, Z.Q. Energy saving of buildings for reducing carbon dioxide emissions using novel dendrite net integrated adaptive mean square gradient. Appl. Energy. 2022, 309, 118409.
- Yang, T.L.; Dong, Q.Y.; Du, Q.Y.; Du, M.; Dong, R.; Chen, M. Carbon dioxide emissions and Chinese OFDI: From the perspective of carbon neutrality targets and environmental management of home country. J. Environ. Manage. 2021, 95, 1131120.
- Mostafaeipour, A.; Bidokhti, A.; Fakhrzad, M.-B.; Sadegheih, A.; Mehrjerdi, Y.Z. A new model for the use of renewable electricity to reduce carbon dioxide emissions. Energy. 2022, 238, 121602.
- Bian, J.S.; Zhang, G.Q.; Zhou, G.H. Manufacturer vs. consumer subsidy with green technology investment and environmental concern. Eur. J. Oper. Res. 2020, 287, 831-843.
- Zhao, S.L.; You, Z.Z.; Zhu, Q.H. Quality choice for product recovery considering a trade-in program and third-party remanufacturing competition. Int. J. Prod. Econ, 2021, 240, 108239.
- Ma, P.; Gong, Y.M.; Mirchandani, P. Trade-in for remanufactured products: pricing with double reference effects. Int. J. Prod. Econ, 2020, 230, 107800.
- He, Z.; Wang, D.F.; He, S.G.; Zhang, Y.W.; Dai, A.S. Two-dimensional extended warranty strategy including maintenance level and purchase time: a win-win perspective. Comput. Ind. Eng. 2020, 141, 106294.
- Huang, H.F.; Liu, F.; Zhang, P. To outsource or not to outsource? Warranty service provision strategies considering competition, costs and reliability. Int. J. Prod. Econ, 2021, 242, 108298.
- Liu, P.; Wang, G.J.; Su, P. Optimal replacement strategies for warranty products with multiple failure modes after warranty expiry. Comput. Ind. Eng. 2021, 153, 107040.
- Geng, W.X.; Fan, Y. Emission trading in an imperfectly competitive product market: a comparison of social welfare under mass-and rate-based schemes. Comput. Ind. Eng. 2021, 162, 107761.
- Lu, C.; Li, W.; Gao, S.B. Driving determinants and prospective prediction simulations on carbon emissions peak for China’s heavy chemical industry. J. Clean. Prod. 2020, 251, 119642.
- Sun, Z.R.; Liu, Y.D.; Yu, Y.N. China’s carbon emission peak pre-2030: exploring multi-scenario optimal low-carbon behaviors for China’s regions. J. Clean. Prod. 2019, 231, 963-979.
- It should be clearly shown, on the basis of two or three examples, the regulations that exist in the field of TON, TOR and EWS in individual US, EU, China markets, whether the rest of Asia, whether and what regulations exist or are planned to be implemented. I do not require it to be very broad, but it should be mentioned in the introduction and at the end in the summary.
Authors’ response: Thank you for your suggestion. In the revised manuscript, we have cited some regulation examples of “trade old for new”, “trade old for remanufactured” and extended warranty service in the Introduction (Please refer to pages 1, 2) and Conclusion (Please refer to page 19, 20) sections.
- What is the CSR mentioned in line 103, does this take into account Customer Social Responsibility? How do you envision this contribution? Please add some examples.
Authors’ response: Thank you for your kind reminder. CSR means that Corporate Social Responsibility is the responsibility of enterprises to consumers, society and the environment while creating profits and being responsible to shareholders and employees. This paper does not directly consider the corporate social responsibility, but starts from the perspective of profit, and then analyzed the impact on consumers, the environment and society. Therefore, in the revised manuscript, we have replaced the document with reference [16] to avoid misunderstanding. Please refer to the lines 116-118 in page 3.
- Do you calculate carbon dioxide emissions hidden in the energy produced and used to produce the product? Do you only take the carbon emissions used in the production of the product itself? Line 195
Authors’ response: Thanks to your thoughtful suggestion. This paper analyzed the carbon emissions involved in the production of products, and does not calculate the carbon dioxide emissions hidden in the energy produced. In future research, we will expand the calculation of carbon emissions.
- Have there been consumer opinion surveys, do they want products such as TON and TOR or EWS? Add info about it in the literature review.
Authors’ response: Thank you for your suggestion. In the revised manuscript, we have added the literatures about consumer opinion surveys about purchasing products with TON or TOR with EWS in the Literature review section (Please refer to pages 3. 4).
- How do these proposals relate to the regulations implemented in the world concerning, for example, returnable bottles and recycling fees? Are there such solutions in China? Add information about it in the summary.
Authors’ response: We are grateful for your comment. In the revised manuscript, we have added some examples about regulations implemented in China and USA in the Conclusion section (Please refer to pages 19, 20).
In conclusion, we would like to express our sincere gratitude to you for your constructive comments and suggestions, which have helped us to significantly improve the quality and presentation of our paper.
Reviewer 3 Report
The authors build on the issues of reducing environmental pollution as more enterprises engage in remanufacturing and recycling of products. They use game theory to design an optimal extended warranty service strategy and analyze its environment impact in a remanufactured supply chain, game theory is used to model the competitive relationship between a manufacturer and an E-commerce platform. The results indicate that when the level of substitutability between remanufactured and new products meets a certain rage, new or remanufactured products with extended warranty service strategy can increase the demand of new or remanufactured products, respectively.
Overall I found the paper very interesting and well-positioned. It addresses an interesting and timely topic and believe it will make a great contribution to the literature. I would suggest the authors through to make some minor edits which could perhaps make the paper a bit more citeable and attract a broader readership.
First, I would recommend that you improve the conclusion section a bit and try to link your results to a bit more broad readership. The results are very interesting and could also link well to the growing literature on circular economy. Specifically I would suggest that you try to compare and extend a bit the findings of other related literature such as the work noted below and say how your work perhaps extends on or takes a different angle from that. The use of digital technologies for circular economy is fascinating and I believe you readership would be interested to see how your work links to that stream of research.
Kristoffersen, E., Mikalef, P., Blomsma, F., & Li, J. (2021). Towards a business analytics capability for the circular economy. Technological Forecasting and Social Change, 171, 120957.
Author Response
The authors build on the issues of reducing environmental pollution as more enterprises engage in remanufacturing and recycling of products. They use game theory to design an optimal extended warranty service strategy and analyze its environment impact in a remanufactured supply chain, game theory is used to model the competitive relationship between a manufacturer and an E-commerce platform. The results indicate that when the level of substitutability between remanufactured and new products meets a certain rage, new or remanufactured products with extended warranty service strategy can increase the demand of new or remanufactured products, respectively.
Overall I found the paper very interesting and well-positioned. It addresses an interesting and timely topic and believe it will make a great contribution to the literature. I would suggest the authors through to make some minor edits which could perhaps make the paper a bit more citable and attract a broader readership.
Authors’ response: Thank you for your recognition and appreciation of our research. We are grateful for your endorsement of our research. Please find below our response to your thoughtful comments that have helped to improve the quality and presentation of our article.
- First, I would recommend that you improve the conclusion section a bit and try to link your results to a bit more broad readership. The results are very interesting and could also link well to the growing literature on circular economy. Specifically, I would suggest that you try to compare and extend a bit the findings of other related literature such as the work noted below and say how your work perhaps extends on or takes a different angle from that. The use of digital technologies for circular economy is fascinating and I believe you readership would be interested to see how your work links to that stream of research.
Kristoffersen, E., Mikalef, P., Blomsma, F., & Li, J. (2021). Towards a business analytics capability for the circular economy. Technological Forecasting and Social Change, 171, 120957.
Authors’ response and revision: Thank you for your suggestion. In the revised manuscript, we have revised the literature review and conclusion sections, updated the references, put forward the differences from the current research and expanded the application of the conclusions (Please refer to pages 3, 4, 19, 20). In the whole revised manuscript, we also conduct a more systematic review and additional relevant literature has been added as listed below.
References
- Nash, S.L.; Steurer, R. Climate change acts in Scotland, Austria, Denmark and Sweden: the role of discourse and deliberation. Clim. Policy. 2021, 21(9), 1-12.
- Nash, S.L.; Steurer, R. Taking stock of climate change acts in Europe: living policy processes or symbolic gestures? Clim. Policy. 2019, 19(8), 1052-1065.
- Zhu, X.X.; Ren, M.L.; Wu, G.D.; Pei J.; Pardalos, P.M. Promoting new energy vehicles consumption: the effect of implementing carbon regulation on automobile in China. Comput. Ind. Eng. 2019, 125: 211-226.
- Kristofferse, E.; Blomsma, F.; Mikalef, P.; Li, J.Y. The smart circular economy: a digital-enabled circular strategies framework for manufacturing companies. J. Bus. Res. 2020, 120, 241-261.
- Ye, F.; Ni, D.B.; Li, K.W. Competition between manufacturers and sharing economy platforms: an owner base and sharing utility perspective. Int. J. Prod. Econ. 2021, 234, 108022.
- Liu, C.Y.; Wang, H.; Tang, J.; Chang, C.T.; Liu, Z. Optimal recovery model in a used batteries closed-loop supply chain considering uncertain residual capacity. Transport. Res. E Logist. Transport Rev. 2021, 156, 102516.
- Zheng, Y.Y.; Zhao, Y.X.; Wang, N.G.; Meng, X.G.; Yang, H.L. Financing decision for a remanufacturing supply chain with a capital constrained retailer: a study from the perspective of market uncertainty. Int. J. Prod. Econ. 2022, 245, 108397.
- Zheng, X.X.; Li, D.F.; Liu, Z.; Jia, F.; Lev, B. Willingness-to-cede behavior in sustainable supply chain coordination. Int. J. Prod. Econ. 2021, 240, 108207.
- Nash, S.L.; Steurer, R. From symbolism to substance: what the renewal of the Danish climate change act tells us about the driving forces behind policy change. Environ. Pollut. 2021, doi: 10.1080/096440.2021.1922186.
- Lverson, L.R.; Thompson, F.R.; Matthews, S.; Peters, M.; Prasad, A.; Dijak, W.D.; Fraser, J.; Wang, W.J.; Hanberry, B.; He, H. Multi-model comparison on the effects of climate change on tree species in the eastern U.S.: results from an enhanced niche model and process-based ecosystem and landscape models. Landsc. Ecol. 2017, 32, 1327-1346.
- Wójcik-Jurkiewicz, M.; Czarbecja, M.; Kinelshi, G.; Sadowska, B.; BiliÅ„ska-Reformat, K. Determinants of decarbonization in the transformation of the energy sector: the case of Poland. Energy. 2021, 14, 1217.
- Kristoffersen, E.; Mikalef, P.; Blomsma, F.; Li, J.Y. The effects of business analytics capability on circular economy implementation, resource orchestration capability, and firm performance. Int. J. Prod. Econ. 2021, 239, 108205.
- Liu, B.; Shen, L.J.; Xu, J.Y.; Zhao, X.J. A complimentary extended warranty: profit analysis and pricing strategy. Int. J. Prod. Econ. 2020, 229, 107860.
- Liu, A.J.; Zhang, Y.; Luo, S.H.; Miao, J. Dual-channel global closed-loop supply chain network optimization based on random demand and recovery rate. Int. J. Environ. Res. Public Health. 2020, 17, 8768.
- Kristoffersen, E.; Mikalef, P.; Blomsma, F.; Li, J.Y. Towards a business analytics capability for the circular economy. Technol. Forecast. Soc. Change. 2021, 171, 120957.
- Zhang, A.; Wang, J.X.; Farooque, M.; Wang, Y.L.; Choi, T.M. Multi-dimensional circular supply chain management: a comparative review of the state-of-the-art practices and research. Transp. Res. Part E Logist. Transp. Rev. 2021, 155, 102509.
- Bian, J.S.; Zhao, X. Competitive environmental sourcing strategies in supply chains. Int. J. Prod. Econ. 2020, 230, 107891.
- Ma, L.; Zhang, X.R.; Du, Y.S. Influence mechanism on supplier emission reduction based on a two-level supply chain. Int. J. Environ. Res. Public Health. 2021, 18, 12439.
- Giri, B.C.; Glock, G.H. The bullwhip effect in a manufacturing/remanufacturing supply chain under a price-induced non-standard ARMA (1,1) demand process. Eur. J. Oper. Res. 2021, doi: 10.1016/j.ejor.2021.10.025.
- Luo, R.L.; Zhou, L.; Song, Y.; Fan, T.J. Evaluating the impact of carbon tax policy on manufacturing and remanufacturing decisions in a closed-loop supply chain. Int. J. Prod. Econ. 2022, 245, 108408.
- Zhou, Y.; Xiong, Y.; Jin, M.Y. Less is more: consumer education in a closed-loop supply chain with remanufacturing. Omega. 2021, 101, 102259.
- Yang, F.; Wang, M.M.; Ang, S. Optimal remanufacturing decisions in supply chains considering consumers’’ anticipated regret and power structures. Transp. Res. Part E Logist. Transp. Rev. 2021, 148, 102267.
- Zhang, W.; He, Y. Optimal policies for new and green remanufactured short-life-cycle products considering consumer behavior. J. Clean. Prod. 2019, 214, 483-505.
- Gan, S.S.; Pujawan, N.; Suparno, Widodo, B. Pricing decision for new and remanufactured product in a closed-loop supply chain with separate sales-cahnnel. Int. J. Prod. Econ. 2016, 190, 120-132.
- Hazen, B.; Overstreet, R.E.; Jones-Farmer, L.A.; Field, H.S. The role of ambiguity tolerance in consumer perception of remanufactured products. Int. J. Prod. Econ, 2012, 135(2): 781-790.
- Han, Y.M.; Li, J.Z.; Lou, X.Y.; Fan, C.Y.; Geng, Z.Q. Energy saving of buildings for reducing carbon dioxide emissions using novel dendrite net integrated adaptive mean square gradient. Appl. Energy. 2022, 309, 118409.
- Yang, T.L.; Dong, Q.Y.; Du, Q.Y.; Du, M.; Dong, R.; Chen, M. Carbon dioxide emissions and Chinese OFDI: From the perspective of carbon neutrality targets and environmental management of home country. J. Environ. Manage. 2021, 95, 1131120.
- Mostafaeipour, A.; Bidokhti, A.; Fakhrzad, M.-B.; Sadegheih, A.; Mehrjerdi, Y.Z. A new model for the use of renewable electricity to reduce carbon dioxide emissions. Energy. 2022, 238, 121602.
- Bian, J.S.; Zhang, G.Q.; Zhou, G.H. Manufacturer vs. consumer subsidy with green technology investment and environmental concern. Eur. J. Oper. Res. 2020, 287, 831-843.
- Zhao, S.L.; You, Z.Z.; Zhu, Q.H. Quality choice for product recovery considering a trade-in program and third-party remanufacturing competition. Int. J. Prod. Econ, 2021, 240, 108239.
- Ma, P.; Gong, Y.M.; Mirchandani, P. Trade-in for remanufactured products: pricing with double reference effects. Int. J. Prod. Econ, 2020, 230, 107800.
- He, Z.; Wang, D.F.; He, S.G.; Zhang, Y.W.; Dai, A.S. Two-dimensional extended warranty strategy including maintenance level and purchase time: a win-win perspective. Comput. Ind. Eng. 2020, 141, 106294.
- Huang, H.F.; Liu, F.; Zhang, P. To outsource or not to outsource? Warranty service provision strategies considering competition, costs and reliability. Int. J. Prod. Econ, 2021, 242, 108298.
- Liu, P.; Wang, G.J.; Su, P. Optimal replacement strategies for warranty products with multiple failure modes after warranty expiry. Comput. Ind. Eng. 2021, 153, 107040.
- Geng, W.X.; Fan, Y. Emission trading in an imperfectly competitive product market: a comparison of social welfare under mass-and rate-based schemes. Comput. Ind. Eng. 2021, 162, 107761.
- Lu, C.; Li, W.; Gao, S.B. Driving determinants and prospective prediction simulations on carbon emissions peak for China’s heavy chemical industry. J. Clean. Prod. 2020, 251, 119642.
- Sun, Z.R.; Liu, Y.D.; Yu, Y.N. China’s carbon emission peak pre-2030: exploring multi-scenario optimal low-carbon behaviors for China’s regions. J. Clean. Prod. 2019, 231, 963-979.
In conclusion, we would like to express our sincere gratitude to you for your constructive comments and suggestions, which have helped us to significantly improve the quality and presentation of our paper.
Round 2
Reviewer 2 Report
Very interesting and now complete article. Adding the context of other countries broadened the view on the topic at hand.
Just check carefully the names and titles of the cited items in the bibliography.
Please complete DOI if they exist for each bibliography.
item 16 has errors in the names.
Author Response
Very interesting and now complete article. Adding the context of other countries broadened the view on the topic at hand.
Authors’ response: We are grateful for your endorsement of our research.
- Just check carefully the names and titles of the cited items in the bibliography.
Authors’ response: Thank you for your kind reminder. we have double-checked the whole manuscript and tried our best to make it typo-free. All the mistakes have been corrected, please refer to pages 1, 3, 4, 5, 12, 14, 18, 19, 27 in the revised manuscript.
- Please complete DOI if they exist for each bibliography.
Authors’ response: Thanks to your thoughtful suggestion. In the revised manuscript, we have added the DOI for each literature in the References section (Please refer to pages 26-29).
- item 16 has errors in the names
Authors’ response: We appreciate your careful reading of our manuscript and putting forward this suggestion. It is our fault that we have not carefully checked the manuscript. We have corrected the mistake (Please refer to page 27) and double-checked the manuscript and tried our best to make it typo-free.
This manuscript is a resubmission of an earlier submission. The following is a list of the peer review reports and author responses from that submission.